# Regulation of B-Cell Receptor Signaling and Its Therapeutic Relevance in Aggressive B-Cell Lymphomas

**DOI:** 10.3390/cancers14040860

**Published:** 2022-02-09

**Authors:** Núria Profitós-Pelejà, Juliana Carvalho Santos, Ana Marín-Niebla, Gaël Roué, Marcelo Lima Ribeiro

**Affiliations:** 1Lymphoma Translational Group, Josep Carreras Leukaemia Research Institute (IJC), 08916 Badalona, Spain; nprofitos@carrerasresearch.org (N.P.-P.); jcarvalho@carrerasresearch.org (J.C.S.); mlima@carrerasresearch.org (M.L.R.); 2Department of Hematology, Experimental Hematology, Vall d’Hebron Institute of Oncology (VHIO), Vall d’Hebron Barcelona Hospital Campus, Vall d’Hebron University Hospital, 08035 Barcelona, Spain; ana.marin@vhebron.net; 3Laboratory of Immunopharmacology and Molecular Biology, Sao Francisco University Medical School, Braganca Paulista 12916-900, Brazil

**Keywords:** B-cell non-Hodgkin lymphoma (B-NHL), B-cell receptor (BCR), Bruton’s tyrosine kinase (BTK), spleen tyrosine kinase (SYK), phosphoinositide-3-kinase (PI3K), ibrutinib, acalabrutinib, combination therapies

## Abstract

**Simple Summary:**

Dysregulated B-cell receptor (BCR) signaling is considered a potent contributor to tumor survival in different subtypes of B-cell non-Hodgkin lymphomas (B-NHLs). In the last decade, BCR-targeted therapies have emerged as promising alternative treatment options to standard chemoimmunotherapy. Despite the initial excitement and strong biological rationale, BCR-targeting drugs often fail to produce durable responses. This review will discuss the current understanding of the role of BCR signaling in B-NHLs. In addition, the mechanisms of action of BCR-targeted therapies, and how our actual knowledge supports the development of more specific inhibitors and new, rationally based, combination therapies, will also be discussed.

**Abstract:**

The proliferation and survival signals emanating from the B-cell receptor (BCR) constitute a crucial aspect of mature lymphocyte’s life. Dysregulated BCR signaling is considered a potent contributor to tumor survival in different subtypes of B-cell non-Hodgkin lymphomas (B-NHLs). In the last decade, the emergence of BCR-associated kinases as rational therapeutic targets has led to the development and approval of several small molecule inhibitors targeting either Bruton’s tyrosine kinase (BTK), spleen tyrosine kinase (SYK), or phosphatidylinositol 3 kinase (PI3K), offering alternative treatment options to standard chemoimmunotherapy, and making some of these drugs valuable assets in the anti-lymphoma armamentarium. Despite their initial effectiveness, these precision medicine strategies are limited by primary resistance in aggressive B-cell lymphoma such as diffuse large B-cell lymphoma (DLBCL) and mantle cell lymphoma (MCL), especially in the case of first generation BTK inhibitors. In these patients, BCR-targeting drugs often fail to produce durable responses, and nearly all cases eventually progress with a dismal outcome, due to secondary resistance. This review will discuss our current understanding of the role of antigen-dependent and antigen-independent BCR signaling in DLBCL and MCL and will cover both approved inhibitors and investigational molecules being evaluated in early preclinical studies. We will discuss how the mechanisms of action of these molecules, and their off/on-target effects can influence their effectiveness and lead to toxicity, and how our actual knowledge supports the development of more specific inhibitors and new, rationally based, combination therapies, for the management of MCL and DLBCL patients.

## 1. Introduction

The introduction of massive sequencing approaches together with the development of more physiological preclinical models, has recently allowed a deeper understanding of the relevance of B-cell signaling pathways in the molecular pathogenesis of B-cell non-Hodgkin lymphoma (B-NHL). These advances provided significant insights into the differential response to standard immunotherapeutic regimens across the most aggressive B-NHL subtypes, including diffuse large B-cell lymphoma (DLBCL) and mantle cell lymphoma (MCL), and yielded several promising targets for novel antitumor therapies for these life-threatening diseases. As a result, in the last two decades the treatment landscape of MCL, and to a lesser extent, DLBCL, has expanded from conventional cytotoxic chemotherapies to encompass targeted small-molecule drugs, monoclonal antibodies, antibody-drug conjugates, and cellular therapies. Among these new approaches, a significant proportion of MCLs and DLBCLs have been shown to be addicted to B-cell receptor (BCR) signaling, and/or to its downstream oncogenic pathways such as nuclear factor–κB (NF-κB) and phosphatidylinositol 3-kinase (PI3K). Based on these observations, various specific inhibitors targeting these signaling cascades have been developed and evaluated in the clinics, essentially in patients with relapsed and/or refractory (R/R) disease. In the following chapters, we provide an overview of current understanding on the role of the BCR pathway in lymphomagenesis, with a special focus on Bruton’s tyrosine kinase (BTK) and PI3K downstream signaling. We also summarize the available data on the clinical activity and the potential mechanisms of resistance to BTK and PI3K inhibitors in these diseases, either as monotherapies or in combination with other biological agents.

## 2. Main Characteristics of Aggressive B-Cell Lymphoma

DLBCL and MCL represent clinically and molecularly heterogeneous malignancies of aggressive B-NHL. Thirty to forty percent of all newly diagnosed B-NHL cases in the Western world are DLBCL, the most common B-NHL neoplasm [1,2]. DLBCL subtypes are classified by cell-of-origin, genetic and molecular features. According to the cell-of-origin, DLBCL can be classified as germinal center B-cell-like (GCB) or activated B-cell-like (ABC) subtypes. GCB and ABC subtypes have distinct genomic alterations and clinical outcomes [3]. GCB-DLBCL presents intra-clonal heterogeneity, somatic hypermutation, and CD10 and B-Cell Lymphoma 6 (BCL-6) expression, while ABC-DLBCL has constitutive activity of NF-kB and expression of B-Cell Lymphoma 2 (BCL-2) and Interferon Regulatory Factor 4 (IRF4) [4]. The molecular stratification of DLBCL is defined as double-hit or triple-hit lymphoma depending on *MYC* and *BCL2* and/or *BCL6* rearrangements [5]. Despite of high heterogeneity, these patients usually present a poor prognosis after frontline treatment [6,7]. Several studies based on whole-exome sequencing analysis further identify new genetic DLBCL subtypes according to recurrent mutations in *MYD88*, *CD79*, *NOTCH1*, *NOTCH2*, *EZH2* and *BCL2* genes [4]. Unfortunately, more than half of DLBCL patients experience disease progression after first line treatment and most patients with R/R DLBCL are largely incurable.

MCL is a B-NHL that accounts for up to 7% of malignant lymphomas in Western Europe with an incidence rate of about 2 per 100,000 individuals every year [8]. Different molecular mechanisms drive the disease evolution resulting in a highly heterogeneous clinical behavior. The primary oncogenic event in MCL development is the chromosomal translocation t(11;14)(q13;q32) which leads to the constitutive overexpression of the cell cycle regulator cyclin D1 [9]. The tumors are classified as nodal or conventional MCL, which represents 80–90% of cases and has a worse prognosis, and as non-nodal leukemic MCL, accounting for 10–20% of cases presenting an indolent biological behavior. These two subtypes have different genetic and molecular features. Nodal MCL tumors are characterized by unmutated immunoglobulin heavy chain variable region genes (IGHV) and by the transcription factor sex-determining region Y-box 11 (SOX11) overexpression, while non-nodal MCL typically harbor IGHV mutation and no SOX11 expression [10]. Although the median overall survival (OS) of MCL patients is 5–7 years [11], major advances in the treatment have been achieved in the last few years with the advent of novel targeted agents such as BTK inhibitors (BTKi).

## 3. B-Cell Receptor (BCR) Signaling

### 3.1. Physiological Roles of BCR Signaling

The BCR complex is essential for B-cell development and function. The BCR consists of a transmembrane immunoglobulin complex essential for the antigen binding on the surface of B lymphocytes and plays a key role in immune response, cell growth, adhesion, differentiation, survival, cytoskeletal remodeling, and apoptosis [12,13]. The rearrangement of immunoglobulin heavy chain variable (V), diversity (D), and joining (J) gene segments in the progenitor B cells give rise to the pre-BCR which, in turn, mediate signal cascade for immature B-cell development. In the dark zone germinal center (GC), mature B cells undergo somatic hypermutation leading to the complete BCR affinity maturation. Then, these B cells experience class-switch recombination and become memory B cells or antibody secreting plasma cells. Following a pathogen infection, B cells use the BCRs to recognize and bind to the antigenic epitopes resulting in the secretion of specific antibodies as an early attempt to neutralize the foreign antigen or secreting cytokines to attract other immune cells [14].

The BCR exerts its function through tonic or antigen-dependent BCR signaling pathways. Tonic BCR signaling is essential for the downstream survival signal of resting B cells and is mediated by phosphatidylinositol 3 kinase (PI3K)/AKT/forkhead box class O family member transcription factors (FOXO)/mammalian target of rapamycin (mTOR) pathways [15,16].

The BCRs harbor the Igα (CD79a)/Igβ (CD79b) heterodimer with an immunoreceptor tyrosine-based activation motifs (ITAMs) that are phosphorylated upon antigen-dependent BCR signaling activation [17]. BCR-antigen microclusters are associated with lipid on the plasma membrane leading to phosphorylation of BCR’s ITAMs by Src kinases, which drive the recruitment of downstream signaling pathways such as spleen tyrosine kinase (SYK) and BTK. SYK, in turn, phosphorylates the adaptor proteins as BLNK, SHC, and BCAP, which then recruit other signaling molecules [18]. The BCR signal transduction is further propagated by PI3K, Rho family kinases, mitogen-activated protein (MAP) kinases, and transcription factors, such as NF-κB [19,20,21,22]. On the other hand, the BCR signal transduction can be inhibited through dephosphorylation of BCR components by phosphatases, such as protein tyrosine phosphatase non-receptor type 22 (PTPN22), Protein Tyrosine Phosphatase Non-Receptor Type 6 (PTPN6), and Phosphatase and Tensin Homolog (PTEN) [23,24].

B-cell development, differentiation and maturation, co-stimulatory protein expression, migration, and apoptosis are tightly regulated by BTK upon BCR stimulation [25,26,27]. BTK is a non-receptor cytoplasmic Tec tyrosine kinase protein that harbors a kinase domain, two SRC homology domains (SH2 and SH3), a N-terminal pleckstrin (PH) domain, and a TEC domain. The BCR stimulation leads to a downstream cascade guided by BTK activation through transphosphorylation at Y551 in the catalytic domain, and autophosphorylation at Y223 in the SH3 domain [28,29]. Upon its activation, BTK starts a phosphorylation positive feedback loop with phospholipase C-γ (PLCγ2), which in turn, regulate downstream mediators such as the MAPK pathway and transcription of nuclear factor of activated T-cells (NFAT) [30]. Considering that aberrant activation of BTK has been associated with the pathogenesis of several B-cell malignancies [31], BTK inhibition has been shown to be an attractive therapeutic approach for the treatment of B-cell disorders.

### 3.2. Dysregulated BCR Signaling in Aggressive B-Cell Lymphomas

Genetic alterations that lead to constitutive BCR activation have been associated to B-NHL development. Antigen-dependent immunoglobulin genes rearrangements/chromosomal translocations, chronic active, and tonic BCR signaling deregulation often occur in normal B cells, which may lead to B-cell malignant transformation. Most B-cell lymphomas still express a BCR and share the same survival signals supplied by BCR expression in normal B cells to promote the survival of tumor cells [9].

Antigen’s recognition causes Ig genes somatic hypermutations and chromosomal translocations that may lead to pathological BCR activation (Figure 1). Some oncogenic signaling pathways such as PI3K and NFκB are activated by IgM resulting in survival and proliferation of B cells, while MAPK and ERK signaling pathways are activated by IgG resulting in plasmocytic differentiation [32]. The BCR signaling deregulation has been implicated in several types of lymphomas, such as chronic lymphocytic leukemia/small lymphocytic lymphoma (CLL/SLL), follicular lymphoma (FL), marginal zone lymphoma (MZL), and MCL [33,34,35,36]. Antigen-associated lymphoproliferative disorders may arise from the expansion of reactive B cells, as suggested by autoantigen-dependent BCR activation in ABC-DLBCL, *Helicobacter pylori*-driven gastric MALT lymphoma and hepatitis C-driven splenic MZLs [37,38]. The BCR engagement by self-antigens is evidenced by the presence of BCR clusters on the ABC-DLBCL cell surface as well as the epidemiological studies supporting that several autoimmune diseases are risk factors for the DLBCL development, suggests that B-cells auto reactivity might be implicated in B-cell malignant transformation [39].

It has been shown that ABC-DLBCL displays chronic active BCR signaling resulting in constitutive NF-κB activity [40] (Figure 1). A variety of mechanisms are involved in this aberrant activity, including mutations in *CARD11*, found in approximately 10% of ABC-DLBCL; loss of function mutations in A20 (a NF-κB negative regulator), found in about 24% of ABC-DLBCL; and *ITAM* mutations, present in approximately 20% of ABC-DLBCL, which prevent BCR endocytosis and cause an increase in BCR expression and a decrease in LYN kinase activity [40,41,42]. The adaptor protein myeloid differentiation primary response 88 (MYD88) is critical for NF-ĸB activation downstream of Toll-like receptors (TLRs) and has also been frequently mutated in ABC-DLBCL [43]. The multiprotein super-complex formed by MYD88-TLR9-BCR, characterized by ibrutinib responsiveness, has been shown to co-localize with mTOR on endolysosomes, where it drives pro-survival signaling in ABC-DLBCL through NF-κB and mTOR activation [44]. On the other hand, another recent study based on in vitro DLBCL model showed that a loss of KLHL14, a negative BCR regulator found to be mutated in about 10% of ABC-DLBCL, induces the NF-κB pathway by activating the MYD88-TLR9-BCR super-complex, which confers relative resistance to the first-in-class BTKi ibrutinib [45].

Similarly, constitutive BCR signaling is also associated with the activation of the canonical and non-canonical NF-κB pathways in MCL in both in vitro and in vivo models which is correlated with poor survival [46,47]. Genetic aberrations have also been linked to constitutive BCR-NF-κB signaling in MCL cases. Somatic mutations affecting *BIRC3* and *TRAF2*, negative regulators of the non-canonical NF-kB pathway, are associated with BTKi resistance [46]. SYK overexpression has been reported in the MCL cell lines and in patient-derived samples [48]. Furthermore, gain-of-function mutations in *CARD11* have been described particularly in relapsed MCL more often than at diagnosis [49]. Similar to ABC-DLBCL, the analysis of 29 MCL cases showed that A20 is also frequently deleted (31%) or promoter-methylated (37,5%); however, the expression level of A20 was unrelated to the genomic status of MCL [50].

In contrast to antigen and chronic active BCR signaling, the antigen-independent signal, termed ‘tonic BCR signaling’, is mediated by PI3Kα and PI3Kδ/AKT/mTOR, but not the NF-κB pathway, to promote the proliferation and survival of malignant B cells [30] (Figure 1). Genomic data have shown that GCB-DLBCL lines exclusively use tonic BCR signaling [51]. In normal B cells, SYK regulates PI3K signaling through its interaction with the p85 subunit of PI3K and by the phosphorylation of CD19 (essential for GCB cells) and BCAP (essential for ABC cells) [52]. It has been suggested that tonic BCR signal is also transmitted via SYK to activate the PI3K pathway in GCB-DLBCL. Some evidence show that GCB-DLBCL cells are sensitive to SYK inhibition and that SYK blockage decreases BCR/PI3K/AKT activity, decreases the proliferation and induces apoptosis [53,54]. In mature B cells, BCR-induced activation of PI3K leads to FOXO phosphorylation by AKT causing its transcriptional repression [55]. Based on tonic BCR signaling-dependent DLBCL model, Szydlowski et al. demonstrated that SYK inhibition decreases AKT activity and eventually leads to FOXO1-mediated cell death [56]. Moreover, CRISPR analysis revealed that GCB-DLBCL cells are dependent on the BCR signaling components CD79A, CD79B, LYN, CD19, and CD81 for their survival [44]. Using the same approach, Havranek et al. showed that tonic BCR signaling is tightly regulated by Y188F mutation in *CD79A*, PTEN protein expression and BCR surface density in GCB-DLBCL cells [51]. In aggressive MCL, the constitutive activation of the PI3K/AKT axis has also been associated to impaired PTEN [57], supporting the synergy between the blockage of BTK and the PI3K/mTOR/Akt pathways found in MCL cases.

## 4. Pharmacological Targeting of BCR Upstream Kinases and Its Limitations in MCL and DLBCL

### 4.1. Preclinical Drug Development

Studies targeting BTK have attracted substantial attention due to its crucial role in the BCR pathway and BTKi have shown promising antitumor activities, both in vivo and in vitro models. According to its mechanism of action and their binding affinity and activity, the BTKi are classified into two types: irreversible inhibitors, where they bind to the amino acid residue Cys481 forming a covalent bond; or the reversible inhibitors, which bind to an inactive conformation of the kinase, by accessing the specific SH3 pocket of BTK [58].

The majority of BTKi are irreversible inhibitors, from which the first-generation inhibitor is ibrutinib. In 2010, ibrutinib was demonstrated to have selective toxicity in DLBCL cell lines with chronically active BCR signaling with sub-nanomolar activity (IC_50_ = 0.5 nM), by preventing the BTK autophosphorylation [40] and in vivo data confirmed its potential [59]. Ibrutinib, when combined with bortezomib, also presented a synergism by enhancing apoptosis in DLBCL and MCL cells through AKT and NFκB inactivation [60], while in DLBCL it showed cumulative antitumor effects with other agents, such as enzastaurin [61] or lenalidomide [62]. Together these preclinical studies provided detailed insights into the mechanism of action for the subsequent clinical trials (described below).

The off-target side effects of ibrutinib have led to the development of second-generation inhibitors of which acalabrutinib demonstrated to have less off-target kinases inhibition [63] and also to be more potent in in vitro assays and in vivo canine models of DLBCL and xenograft models derived from ABC-DLBCL and MCL [63,64,65]. Another more selective inhibitor is zanubrutinib, which also showed antitumor activity in the nanomolar range in MCL cell lines as well as in ABC-DLBCL cells, with a similar effect as ibrutinib but with less off-target effects and prolonged overall survival in a DLBCL xenograft model [66,67]. Alternative irreversible BTKi are M2951 and M7583, presenting an in vivo antitumor activity in preclinical models of ABC-DLBCL [68]; spebrutinib (CC-292), with high efficacy as a single agent as well as in synergistic combinations in ABC-DLBCL, but limited efficacy in GCB-DLBCL [69]; tirabrutinib (ONO/GS-4059), which is showing promising results in preclinical studies [70,71]; TG-1701, a more selective inhibitor, presenting Ikaros as an important biomarker for response and efficacy, both in vitro and in vivo [72]; and other compounds, currently under development with promising results in vitro, such as QL47 [73].

However, primary and secondary resistances have emerged from the irreversible BTKi ibrutinib, resulting in a poorer prognosis of relapsed lymphoma patients. In DLBCL and MCL the mechanisms of resistance are not as well-known as in CLL, where the cause of resistance has been identified to be related in part to mutations at the covalent site (C481) on BTK and also PLCγ2, resulting in a downstream signal activation [74].

To overcome the irreversible BTKi limitations (off-target side effects and long-term toxicities), new reversible compounds are under development that will be helpful for the ibrutinib-resistant patients. Some of the most relevant reversible inhibitors are ARQ-531, which have demonstrated potent activity against both BTK wild-type and C481S mutant in ABC- and GCB-DLBCL cell lines [75,76]; LFM-A13, a dual inhibitor against BTK and PLK [77]; HBW-3-10, with great potency and pharmacokinetic profile and better results than the ibrutinib, when compared in xenograft models [78]; fenebrutinib, a potent reversible inhibitor against BTK C4815 mutant [79], CG-806, with great effects in MCL and DLBCL in vitro and MCL PDX mice models [80], CB1763, with a strong effect on C481S mutant BTK [81]; and Pirtobrutinib (LOXO-305), with potent antitumor effects both in vitro and in vivo [82,83].

Another interesting novel strategy to overcome ibrutinib resistance is the proteolysis-targeting chimera (PROTAC), which will selectively target BTK C481S mutant. L18I has been reported to inhibit proliferation in BTK mutant DLBCL cell lines and induce rapid tumor regression in C481S BTK HBL-1 xenograft tumors [84]. In addition, P13I and compound 6e, are under development in preclinical studies with promising results [85,86,87,88].

The next key component of the BCR signaling pathway, which has gained relevance as a therapeutic target, is SYK. R406 is a SYK inhibitor that induces apoptosis in the majority of DLBCL cell lines studied by the inhibition of both tonic and induced BCR signaling [53]; however, it also inhibits other kinases [89]. Its oral prodrug is R788, fostamatinib, which has shown great efficiency in B-NHL-like mice models and is currently in clinical trials (see below) [90,91]. Another SYK inhibitor is cerdulatinib, which has demonstrated great antitumor activity by inducing apoptosis and cell cycle arrest in both ABC- and GCB-DLBCL cell lines [92]. Similar results were being observed for PRT060318, where the cell cycle arrest effect by the inhibitor was mimicked by a genetic reduction of SYK using a siRNA [89]. Entospletinib (GS-9973) selectively inhibits SYK and presents a synergistic antitumor effect with vincristine, both in a panel of DLBCL cell lines and in a DLBCL tumor xenograft model [93]. TAK-659 is a dual SYK/FLT-3 inhibitor with antitumor activity in DLBCL cell line xenograft models and in patient-derived xenografts (PDX) [94,95]. ASN002 is a dual JAK/SYK inhibitor, showing a great antiproliferative activity in in vitro and in in vivo models. Moreover, it also showed antitumor activity in ibrutinib-resistant cell lines [96].

Another strategy, which has shown effective preclinical results is the inhibition of PI3K pathway, usually in combination with other treatments. These inhibitors can be divided into non-selective pan inhibitors, targeting all isoforms, or isoform-specific inhibitors, which are more selective and have fewer off-target effects [97]. There are several pan inhibitors, of which we highlight buparlisib, with a subnanomolar activity against PI3Kα, β, γ, and δ [98]; copanlisib, a second-generation inhibitor against the α and δ isoforms [99,100]; KA2237, against the α and β isoforms and with tumorigenic properties in both sensitive- and ibrutinib-resistant cells in MCL cell lines and PDX tumors [101]; and tenalisib (RP6530), a dual PI3K δ/γ inhibitor with an enhanced antitumor response by targeting both lymphoma cells and its microenvironment [102]. In addition, there are isoform-specific inhibitors for PI3Kα, such as taselisib (GDC-0032) with a synergistic antiproliferation activity in DLBCL cell lines when combined with venetoclax [103]; alpelisib, with a marginal response when used as a single agent [104] and BYL719, which combined with idelalisib cooperates and inhibitors the proliferation in ABC-DLBCL cell lines [105]. Regarding PI3Kβ inhibitors, preclinical studies using DLBCL models treated with AZD6482, show a low antitumor activity as a single agent [104]. The PI3Kγ inhibitor JN-PK1 is currently under preclinical evaluation, showing inhibition of the proliferation in several lymphoma cell lines [106]. Finally, the predominant isoform in FL and DLBCL, and therefore the most studied for its inhibition is the PI3Kδ. From here, we will highlight several inhibitors such as idelalisib (CAL-101), which downregulated p-AKT and increased the expression of apoptosis markers in DLBCL, MCL, and FL cell lines [107]; duvelisib (IPI-145), with preclinical results showing an inhibition on tumor growth both in vitro and in PDX mice in MCL and synergism with other treatments [108,109]; umbralisib (TGR-1202), a PI3Kδ/CK1 inhibitor that has a synergistic cytotoxic effect when combined with a NF-kB inhibitor (carfilzomib) in MCL cell lines and also when combined with ublituximab (Anti-CD20 antibody) and TG-1801 (bi-specific anti-CD19/CD47 antibody) [110,111]; AMG319 which as a monotherapy has antitumor activity [50] and in combination with vincristine synergistically reduced proliferation in vitro and enhanced tumor regression in vivo in DLBCL [51]; and SHC014748M that is a novel compound with in vitro and in vivo efficacy in B-cell lymphoma cell lines [112].

Lastly, elevated or aberrant activation of mTOR has been identified in several cell lines and patient samples of DLBCL and MCL; thus, its targeting is a therapeutic approach alone or in combination [113]. Rapamycin, or sirolimus, is an antibiotic and the first mTOR inhibitor discovered, from which novel rapamycin analogs have emerged: temsirolimus that has shown antitumor activity in MCL cell lines [114]; and everolimus (RAD001), showing an effect associated with cell-cycle arrest in MCL [115]. To achieve a more potent anticancer activity, small molecules that inhibit both mTORC1 and mTORC2 have been developed. Among them, CC-223 is a potent and selective inhibitor that shows a better induction in apoptosis, when compared to rapamycin in a panel of DLBCL, FL, and MCL cell lines [116]. AZD014 has been shown to synergize with ibrutinib and cause tumor regression in in vivo experiments in ABC-DLBCL [117]. Finally, PQR309 is a dual PI3K/mTOR inhibitor with promise for advancing into clinical trials, alone or in combination with other treatments [118]. Figure 1 summarizes the main BCR signaling therapeutic targets in DLBCL and MCL.

### 4.2. Clinical Experience with the Targeting of Apical BCR Kinases in DLBCL and MCL

The arrival of ibrutinib to the therapeutic armamentarium of MCL was a real breakthrough and has changed the treatment paradigm in the relapsed/refractory (R/R) setting. The first clinical results from the phase 2 pivotal study in heavily pre-treated MCL [119] led to an accelerated approval of ibrutinib for R/R MCL in 2013 by the U.S. Food and Drug Administration (FDA) and in 2014 by the European Medicines Agency (EMA). Ibrutinib showed an overall response (OR) and complete response (CR) rates of 68% and 21%, respectively, with a median duration of response (DOR) of 17.5 months, which were significantly higher than those observed with the previously approved targeted agents, bortezomib, temsirolimus, and lenalidomide, whose rates of OR and CR were 22–33% and 8–10%, respectively, with median DOR ranging between 8 months with bortezomib [120] and temsirolimus [121,122,123] and 16 months with lenalidomide [124,125]. The phase 3 RAY study demonstrated the superiority of ibrutinib versus temsirolimus also in terms of progression free survival (PFS) (15.6 vs. 6.2 months) [126]. In addition, patients receiving ibrutinib on their first relapse showed a significantly higher PFS than those treated with ibrutinib in later relapses (25.4 vs. 10.3 months, respectively), which was an unexpected finding, not observed in the temsirolimus arm, where the PFS was the same for all the patients regardless of the number of previous lines received. Another study pooling together the data from up to 330 patients with R/R MCL prospectively included in two phase 2 trials and in the phase 3 RAY study confirmed that the sooner ibrutinib is used when patients with MCL relapse after the first line, the better the results will be in terms of PFS, but also of CR rate (32% when ibrutinib was used after only one previous line versus 14% when used in later relapses), DOR (35.6 vs. 16.6 months with early vs. late use of ibrutinib) and overall survival (OS) (61.6 vs. 22.5 months) [127], leading to the current unanimous recommendation about the preferential use of ibrutinib as the first option in relapse, in order to get the most out of this active drug in R/R MCL.

Ibrutinib is a well-tolerated and safe drug, but the concern about its collateral effects on other kinases besides BTK, associated with a higher risk of bleeding (mostly grade 1–2 events in the clinical practice) and cardiovascular events (atrial fibrillation and hypertension) led to the development of new generation BTKi with reduced off-target effects. Acalabrutinib and zanubrutinib are the first two of this new generation BTKi with an improved safety profile [128,129] approved for patients with R/R MCL, with encouraging data on efficacy and survival confirming how this family of drugs has definitely changed the landscape of salvage treatment in MCL. Indeed, several newer, safer, and more potent BTKi are coming along, both covalent irreversible (such as acalabrutinib and zanubrutinib) and non-covalent reversible BTKi (such as pirtobrutinib and others), whose main clinical data are summarized in Table 1. In addition, recruiting information and preliminary results of clinical trials with targeted BCR inhibition as a single agent or combinatorial treatments are presented in Appendix A.

The clinical experience with PI3K inhibitors (PI3Ki) in MCL is far from that of BTKi and none of the PI3Ki tested in aggressive B-NHL (MCL or DLBCL), either δ-selective or pan-PI3Ki, has been approved to date, mainly due to an insufficient efficacy in monotherapy. One of the most recently evaluated PI3Ki in MCL and DLBCL, parsaclisib, a δ-selective PI3Ki with an improved toxicity profile, showed initial promising results in a phase 1/2 study [149], which unfortunately were not confirmed later on in the respective phase 2 studies. The CITADEL-205 study in R/R MCL showed activity of parsaclisib only in patients who had not received previous BTKi, with an objective response rate (ORR) and CR of 70% and 15.6%, respectively [150], but the results in the most realistic cohort, the BTKi-experienced patients, were disappointing, with ORR and CR of only 35.5% and 2.9%, respectively [151]. In R/R DLBCL, despite an initial ORR of 30% in the phase 1/2 study [149], the CITADEL-202 study was prematurely closed due to the high proportion of patients with disease progression during treatment: 95% (52 out of 55 patients) in the BTKi-naïve arm and 80% (4 out of 5 patients) in the BTKi-experienced arm [144]. These and other PI3Ki are currently undergoing evaluation in combination with chemotherapy and other targeted drugs (Table 2).

Inhibitors of mTOR (mTORi) and SYK (SYKi) have also been evaluated in MCL and DLBCL. The mTORi temsirolimus was approved in 2009 in Europe for R/R MCL based on its efficacy in two phase 2 studies, showing ORR of 38-41% in monotherapy [121,122], later confirmed in phase 3 studies, with ORR between 22-47% [123,126]. Regarding SYKi, fostamatinib showed modest activity in monotherapy in both MCL and DLBCL in a phase 1/2 study, with ORR of 11% and 22%, respectively [148], and, more recently, TAK-659, a dual inhibitor of SYK and FMS-like tyrosine kinase 3 (FLT3), showed ORR of 28% in patients with R/R DLBCL [152]. Studies evaluating these and other inhibitors of mTOR and SYK, both in monotherapy and in combination, are summarized in Table 1 and Table 2**.**

## 5. BTKi- and PI3Ki-Based Combination Therapies in DLBCL and MCL

In R/R DLBCL, the results of efficacy with BTKi in monotherapy have been modest, with ORR of 38% in ABC-DLBCL (5% in GCB-DLBCL) with ibrutinib [134] or 35% with tirabrutinib, another new generation covalent BTKi [175]. Nevertheless, these results suggested potential benefit in combination, and there are currently several studies ongoing evaluating BTKi with chemotherapy and other targeted agents in DLBCL and other B-cell malignancies (Table 1). Indeed, preclinical studies showed potential synergy and potentially non-overlapping toxicity of the dual inhibition of BCR signaling key kinases, BTK and PI3K [176].

The first attempt to combine BTKi with PI3Ki in a high-risk MCL population was recently published (NCT02268851) [166]. The authors showed that ibrutinib in combination with umbralisib is well tolerated with an expectable and manageable toxicity in a set of 21 MCL patients. Regarding hematological toxicities, cytopenias were frequent and included: neutropenia (8/21 (38%)), thrombocytopenia (10/21 (48%)), and anemia (9/21 (43%)). The non-hematological adverse events (AEs) included diarrhea (14/21 (67%)), infection (7/21 (33%)), nausea (9/21 (43%)), fatigue (11/21 (52%)), and transaminitis (6/21 (29%)). In the efficacy analysis, the authors observed an ORR of 67% (14/21) and the CR rate of 19% (4/21). Although the doublet was well tolerated, the efficiency data were similar to those previously reported with ibrutinib as monotherapy [166]. Considering the lack of efficacy using the double combination, the authors also explored the ibrutinib–umbralisib duet in combination with the type II CD20 antibody ublituximab in R/R B-NHL patients (NCT02006485) [177]. In this study, 46 patients were enrolled (n = 6 DLBCL and n = 6 MCL). In general, the triplet therapy was well tolerated, and the most frequent adverse effects were diarrhea (59%), fatigue (50%), infusion-related reaction (43%), dizziness (37%), nausea (37%), cough (35%), insomnia (33%), neutropenia (33%), pyrexia (33%), thrombocytopenia (28%), and peripheral oedema (26%). In MCL patients, 3/6 (50%) reach CR and 3/6 (50%) partial response (PR). Among DLCBL patients, only 1/6 (17%) has PR while 5/6 (83%) progressive disease. These data showed the tolerable safety profile of the triplet regimen in MCL and DLBCL; however the highly proliferative nature of these tumors might be responsible for the lack of meaningful responses [177]. The main limitation of this study is the small set of patients; thus, caution should be applied to make any conclusions in this population. The BTKi (TG1701)-PI3Ki (umbralisib)-antiCD20 (ublituximab) triplet combination was also evaluated in a phase 1 parallel dose-escalation study (NCT03671590). Preliminary data indicated that the combination is well tolerated and shows enhanced depth of response over TG-1701 monotherapy [178,179]. Recruitment to this study (NCT03671590) continues.

Acalabrutinib was also studied in combination with PI3K inhibitor ACP-319 and was shown to be safe and tolerable in early phase clinical trials in R/R B-cell malignancies, including DLBCL and MCL (NCT02328014) [180]. Additionally, a phase 1b/2 study is underway looking at the combination of acalabrutinib and duvelisib in R/R indolent NHLs (NCT04836832). Furthermore, idelalisib is also under evaluation in combination with ibrutinib and pembrolizumab (anti-PD1) in patients with low-grade B-cell NHLs in a phase II trial (NCT02332980). Parsaclisib is also being evaluated in combination with rituximab, rituximab plus bendamustine, and ibrutinib (NCT03424122). Future translational science will elucidate the mechanistic basis of the outcomes of combinatory BTK and PI3K inhibition to optimize the landscape of B-NHL therapy.

## 6. Conclusions and Perspectives

With the development of genomic sequencing and immunotherapy, the treatment of aggressive B-cell lymphoma entered the era of precision therapy almost two decades ago. Targeting BCR signaling with oral kinase inhibitors has changed the treatment landscape in MCL. In this disease, the introduction of BTK inhibitors represents one of the most important advances in small molecule-based therapies, with high response rates and durable remissions in the relapse setting, and very encouraging results in the frontline setting. Conversely, despite the initial excitement and strong biological rationale, the latter has yet to demonstrate significant improvements in the outcome for DLBCL patients. As an example, single-agent ibrutinib is active and well tolerated in non-GCB DLBCL, but the duration of response is remarkably short in unselected patients, compared to MCL. In addition, in DLBCL, these approaches are often associated with significant toxicities in combination with chemo-immunotherapeutic regimens, such as R-CHOP, probably underlying the biological complexity and aggressiveness of this disease. Thus, durable responses in DLBCL patients will undoubtedly require combination therapies targeting genetically and experimentally validated biologic drivers. In this sense, “chemo-free” combination strategies associating ibrutinib with rituximab and lenalidomide in previously untreated non-GCB cases are showing prolonged responses with some complete responses, suggesting that these new regimens may have a role in frontline therapy. However, longer follow-ups will be required to confirm these promising results.

Besides DLBCL, these combinations may also improve the clinical outcomes of MCL patients, while ensuring manageable toxicity. The most recent trials are taking advantage of the safety profile of second-generation PI3Ki or BTKi, associating them either with secondary agents with a distinct mechanism of action, such as the BCL-2 antagonist venetoclax, the cyclin-dependent kinase (CDK)4/6 inhibitor (palbociclib), or together (ibrutinib + umbralisib or ibrutinib + copanlisib), in order to enhance the blockade of BCR signaling.

Finally, the use of CAR-T and immune checkpoint blockade therapies, such as programmed cell death protein 1 (PD1)/PD1 ligand (PD-L1) antagonists in combination with BTK (and probably PI3K) inhibitors may represent a significant step towards tailored medicine for the clinical management of both MCL and DLBCL.

## Figures and Tables

**Figure 1 cancers-14-00860-f001:**
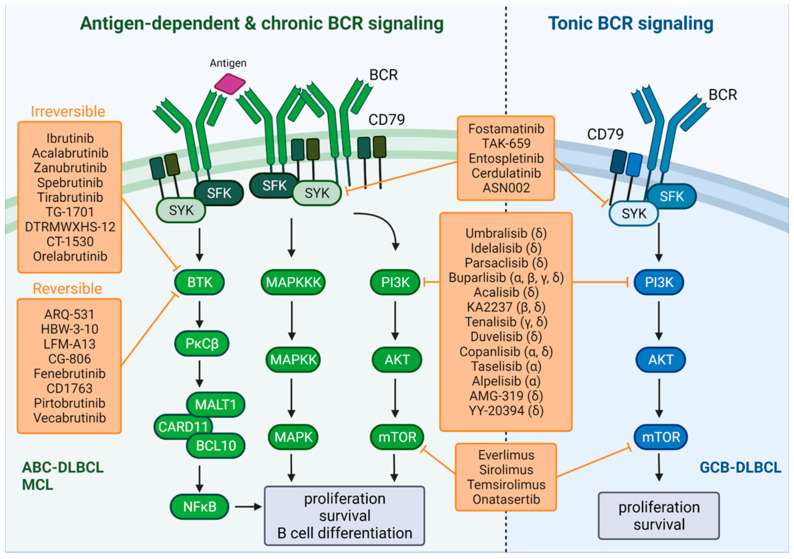
Regulation of BCR signaling and the therapeutic inhibition of BTK and PI3K in DLBCL and MCL. (**Left** panel) Antigen-dependent and chronic active BCR signaling; (**Right** panel) Tonic BCR signaling.

**Table 1 cancers-14-00860-t001:** Clinical trials with targeted BCR inhibition as single agent treatments.

Targets	Drug/Regimen	Clinical Trial	Phase	Nb Pts	Status	Conditions	Response Data	References
BTK	Acalabrutinib	NCT02112526	1	21	Active	R/R DLBCL	ORR 24%, CR 19% AEs Grade 3/4 44%	[130]
BTK	DTRMWXHS-12	NCT02891590	1	13	Completed	R/R B-cell Lymphomas	Well-tolerated and no DLT achieved	[131]
BTK	Ibrutinib	NCT00849654	1	66	Completed	B-cell Lymphomas	ORR 60% CR 16% PFS 13.6 months	[132]
BTK	Ibrutinib	NCT01704963	1	15	Completed	R/R B-cell Lymphomas	ORR 73.3%	[133]
BTK	Ibrutinib	NCT01325701	2	78	Completed	R/R DLBCL	CR or PR in 37% (ABC) and in 5% (GCB)	[134]
BTK	Ibrutinib	NCT02207062	2	20	Active	R/R B-cell Lymphomas	ORR 35% CR 15% PFS 4.1 months OS 22.8 months	[135]
BTK	Ibrutinib	NCT01804686	3	700	Active	CLL, SLL, MCL, FL DLBCL, WM	CR 27.6% PR 42.2% PFS 12.5 months	[136]
BTK	TG1701	NCT03664297	1	86	Active	B-cell Lymphomas	NA	NA
BTK	Vecabrutinib	NCT03037645	1 & 2	39	Terminated	CLL, SLL, MCL, WM, DLBCL, FL, MZL	Well tolerated but terminated due to insufficient evidence of activity	NA
BTK	Zanubrutinib	NCT03189524	1	44	Completed	R/R MCL	CR 86.6% DOR 19.5 months PFS 22.1 months	[129]
BTK	Zanubrutinib	NCT03145064	2	41	Completed	DLBCL	ORR 29.3% CR 17.1% DOR 4.5 months PFS 2.8 months?	[137]
mTOR	Onatasertib	NCT01177397	1 & 2	173	Completed	MM, DLBCL	Acceptable safety PR 17.6%	[138]
PI3K	Acalisib	NCT01705847	1	39	Completed	B-cell Lymphomas	ORR 28.6% AEs grade > 3 55.3%	[139]
PI3K	AMG-319	NCT01300026	1	28	Completed	CLL, DLBCL, MCL	AEs grade > 3 25%	[140]
PI3K	Buparlisib	NCT01693614	2	72	Completed	DLBCL, MCL, FL	ORR 11.5% in DLBCL and 22.7% in MCL	[141]
PI3K	Buparlisib	NCT01719250	Early 1	7	Completed	R/R DLBCL, R/R FL, R/R MCL	NA	NA
PI3K	Fimepinostat	NCT01742988	1	106	Completed	R/R DLBCL	CR 12.5% PR 37.5% SD 37.5%	[142]
PI3K	Idelalisib	NCT03151057	1	60	Active	CLL, FL, MCL, DLBCL	NA	NA
PI3K	KA2237	NCT02679196	1	23	Completed	B-cell Lymphomas	ORR 37% AEs grade > 3 43%	[143]
PI3K	Parsaclisib	NCT03688152	1	9	Completed	R/R DLBCL	NA	NA
PI3K	Parsaclisib	NCT03314922	1	17	Active	B-cell Lymphomas	NA	NA
PI3K	Parsaclisib	NCT02998476	2	60	Completed	R/R DLBCL	ORR 25.5% DOR 6.2 months	[144]
PI3K	Tenalisib	NCT02017613	1	35	Completed	B-cell Lymphomas	ORR 19% CR 6% PR 13%	[145]
PI3K	Umbralisib	NCT01767766	1	90	Completed	NHL, CLL	ORR 24% CR 8% PR 16% AEs grade > 3 in less than 5%	[146,147]
SYK	Fostamatinib	NCT00446095	1 & 2	81	Completed	B-cell Lymphomas	ORR 22% in DLBCL and 11% in MCL PFS 4.2 months	[148]

Abbreviations: FL, follicular lymphoma; DLBCL, diffuse large B-cell lymphoma; MCL, mantle cell lymphoma; MZL, marginal zone lymphoma; NHL, non-Hodgkin lymphoma; CLL, chronic lymphocytic leukemia; MM, multiple myeloma; WM, waldenstrom’s macroglobulinemia; ORR, overall response rate; CR, complete response; PR, partial response; SD, stable disease; PD, progressive disease; DOR, duration of response; PFS, progression-free survival; OS, overall survival; AEs, adverse effects; NA, not available.

**Table 2 cancers-14-00860-t002:** Clinical trials with targeted BCR inhibition in combinatorial treatments.

Targets	Drug/Regimen	Clinical Trial	Phase	Nb Pts	Status	Conditions	Response Data	References
BTK PD1	Acalabrutinib + Pembrolizumab	NCT02362035	1 & 2	161	Active	R/R DLBCL	ORR 26% Discontinuation was due to PD (62%) and AEs (26%)	[153]
BTK	Acalabrutinib + R-CHOP	NCT03571308	1 & 2	39	Active	nHL	NA	NA
BTK	Ibrutinib + R-CHOP	NCT01855750	3	838	Completed	B-cell Lymphomas	ORR 93.6%	[154]
BTK	Ibrutinib + R-ICE	NCT02219737	1	26	Completed	DLBCL	ORR 90%	[155]
BTK	Ibrutinib + CAR-T cell	NCT05020392	3	24	Active	DLBCL, MCL, CLL, SLL, BL	ORR 83%	[156]
BTK PDL1 4-1BB CD20	Ibrutinib + Avelumab + Utomilumab + Rituximab	NCT03440567	1	16	Active	R/R DLBCL, R/R MCL, Transformed FL	NA	NA
BTK	Ibrutinib + Immuno-chemotherapy	NCT02055924	1	85	Terminated	B-cell Lymphomas	CR 42% PR 25% Terminated due to due to veno occlusive disease	[157]
BTK JAK1	Ibrutinib + Itacitinib	NCT02760485	1 & 2	33	Active	B-cell Lymphomas	ORR 24%	[158]
BTK	Ibrutinib + Lenalidomide	NCT01955499	1	34	Active	R/R DLBCL, R/R FL, R/R MZL, R/R MCL	NA	NA
BTK CD20	Ibrutinib + Rituximab	NCT01980654	2	80	Completed	B-cell Lymphomas	ORR 85–75%	[159]
BTK CD20	Ibrutinib + Rituximab + Bendamustine	NCT01479842	1	48	Active	MZL, FL, MCL, WM	OR 94% in MCL and 37% in DLBCL CR 76% in MCL and 31% in DLBCL	[160]
BTK	Ibrutinib + Rituximab + Lenalidomide	NCT02636322	2	60	Active	DLBCL	ORR 65% DOR 15.9 months	[161]
BTK CD20	Ibrutinib + Rituximab + Lenalidomide	NCT02077166	1 & 2	134	Completed	R/R DLBCL	ORR 47% CR 28% PFS 21 months AEs grade > 3 in less 30% patients	[162]
BTK CD20	Ibrutinib + Rituximab + Venetoclax	NCT03136497	1	10	Active	R/R DLBCL	NA	NA
BTK	Spebrutinib	NCT01351935	1	113	Completed	B-cell Lymphomas	ORR 53%	[163]
BTK	Spebrutinib + Lenalidomide	NCT01766583	1	18	Completed	R/R B-cell Lymphomas	NA	NA
BTK CD20	Zanubrutinib + Rituximab	NCT03520920	2	41	Completed	MZL, FL, DLBCL	ORR 35% PFS 3.38 months	[164]
BTK mTOR	DTRMWXHS-12 + Everolimus + Pomalidomide	NCT02900716	1	48	Completed	B-cell Lymphomas	Well-tolerated and no DLT achieved	[131]
BTK PI3K	Ibrutinib + Umbralisib	NCT02874404	2	13	Completed	R/R DLBCL	ORR 31% PFS 3 months	[165]
BTK PI3K CD20	Ibrutinib + Parsaclisib+ Rituximab+ Bendamustine	NCT03424122	1	50	Active	B-cell Lymphomas	NA	NA
BTK PI3K	Ibrutinib + Umbralisib	NCT02268851	1	45	Active	CLL, SLL, MCL	ORR 67% CR 19% PR 48% AEs grade >3 in less than 10%	[166]
BTK PI3K CD20	Ibrutinib + Umbralisib + Ublituximab + Bendamustine	NCT02006485	1	160	Completed	B-cell Lymphomas	DOR 20 months	[167]
mTOR	Everolimus + Lenalidomide	NCT01075321	1 & 2	58	Completed	MZL, FL, MCL, WM	ORR 27%	[168]
mTOR	Everolimus + Panobinostat	NCT00962507	1	11	Completed	B-cell Lymphomas	ORR 43% CR 15%	[169]
mTOR	Everolimus + Panobinostat	NCT00978432	2	50	Terminated	DLBCL	Terminated due to toxicities, which seemed to outweigh the benefits	[170]
mTOR	Everolimus + Panobinostat	NCT00918333	1 & 2	124	Completed	MZL, BL, MCL, SLL, CLL, ALL, WM	NA	NA
mTOR CD20	Everolimus + Rituximab	NCT00869999	2	26	Completed	DLBCL	OR 38% SD 8% DOR 8.1 months	[171]
mTOR	Everolimus + Sorafenib	NCT00474929	1 & 2	103	Completed	B-cell Lymphomas	ORR 30% in DLBCL and 38% in MCL DOR 5.7 months	[172]
mTOR	Everolimus + Sotrastaurin	NCT01854606	1	31	Completed	ABC DLBCL	Due to suboptimal tolerability of the combinations the phase II is not conducted	NA
mTOR	Sirolimus + hyperCVAD	NCT01184885	Early 1	7	Completed	ALL, BL, MCL NA	NA	NA
mTOR CD22	Temsirolimus + Inotuzumab oxogamicin	NCT01535989	1	25	Completed	R/R B-cell Lymphomas	PR 39% This drug combination is not possible due to toxicities	[173]
PI3K CD20	Buparlisib + Rituximab	NCT02049541	1	18	Active	R/R FL, R/R MZL, R/R MCL, WM	NA	NA
PI3K SYK	Idelalisib + Entospletinib	NCT01796470	2	66	Terminated	B-cell Lymphomas	Terminated due to pneumonitis in 18% of patients	[174]
SYK	TAK-659 + R-CHOP	NCT03742258	1	12	Active	DLBCL	NA	NA

Abbreviations: FL, follicular lymphoma; DLBCL, diffuse large B-cell lymphoma; MCL, mantle cell lymphoma; MZL, marginal zone lymphoma; NHL, non-Hodgkin lymphoma; CLL, chronic lymphocytic leukemia; MM, multiple myeloma; WM, waldenstrom’s macroglobulinemia; ORR, overall response rate; CR, complete response; PR, partial response; SD, stable disease; PD, progressive disease; DOR, duration of response; PFS, progression-free survival; OS, overall survival; AEs, adverse effects; NA, not available.

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
