# Peer review of "Regulation of B-Cell Receptor Signaling and Its Therapeutic Relevance in Aggressive B-Cell Lymphomas"

_cancers, 2022, doi:10.3390/cancers14040860_

Round 1

Reviewer 1 Report

The review by Profitos-Peleja et al. on BCR signaling and interest of different inhibitors of BCR signalling in the treatment of B-cell lymphomas is well written, clear, complete, and informative, and therefore deserves publication in Cancers.

Author Response

We gratefully acknowledge the positive valuation of our work made by the reviewer.

Reviewer 2 Report

An extensive review of work done by the authors is commendable. I have few minor suggestions which I hope the authors can address them. 

Page 2 line 95,96. Seems like there is a missing word after binding. “that serve as antigen binding ……”

Page 8 line 362. Authors should consider removing “ibrutinib” and leave zanubrutinib and acalabrutinib.

Table 1 should be divided to include single and combinatorial therapies separately. It will be easy to follow in that case. Also they should remove the list of studies with no reports, recruiting and yet to be recruiting to a supplementary table.

Author Response

An extensive review of work done by the authors is commendable. I have few minor suggestions which I hope the authors can address them.

Comment 1)  Page 2 line 95,96. Seems like there is a missing word after binding. “that serve as antigen binding ……”

Answer)  We thank the reviewer for annotating this mistake. The text has been corrected in the new version f the manuscript (lines 112-113):

“The BCR consists of transmembrane immunoglobulin complex essential for the antigen binding on the surface of B lymphocytes and plays a key role in immune response, cell growth, adhesion, differentiation, survival, cytoskeletal remodeling and apoptosis.”

Comment 2) Page 8 line 362. Authors should consider removing “ibrutinib” and leave zanubrutinib and acalabrutinib.

Answer)  We followed the recommendation of the reviewer and modified the text as follows (lines 371-372):

“Indeed, several newer, safer, and more potent BTKi are coming along, both covalent irreversible (like acalabrutinib and zanubrutinib) and non-covalent reversible BTKi (such as pirtobrutinib and others)”

Comment 3) Table 1 should be divided to include single and combinatorial therapies separately. It will be easy to follow in that case. Also they should remove the list of studies with no reports, recruiting and yet to be recruiting to a supplementary table.

Answer)  On the request of the reviewer, we have split Table 1 in two tables, with the Table 1 describing the clinical trials using BTKi single agents, and the new Table 2 centered on combination studies. In addition, the most recent trial with no preliminary data  available, have been included in a new Supplemental Table S1.

The text was modified accordingly:

Lines 373-375: “In addition, recruiting information and preliminary results of clinical trials with targeted-BCR inhibition as single agent or combinatorial treatments are presented in Supplemental Table S1.”

Table 1 title (line 376): “Table 1. Clinical trials with targeted-BCR inhibition as single agent treatments”

Lines 396-398: “These and other PI3Ki are currently undergoing evaluation in combination with chemotherapy and other targeted drugs (Table 2).”

Lines 406-407: “Studies evaluating these and other inhibitors of mTOR and SYK, both in monotherapy and in combination, are summarized in Table 1 and 2.”

Table 2 title (line 409): “Table 2. Clinical trials with targeted-BCR inhibition in combinatorial treatments.”

Reviewer 3 Report

The authors report in a well-organized way the current developments in precision therapy with the use of inhibitors of BTK, SYK and PI3K, providing a clear view of the efficacy, side effects and future prospects of new therapeutic strategies.

Author Response

(The authors gave the same response as above.)

Reviewer 4 Report

This review manuscript is highlighting a general overview of a current understanding of the role of the BCR pathway in lymphomagenesis, with a special focus on Bruton’s tyrosine kinase (BTK) and PI3K downstream signalling. The authors also summarize the available data on the clinical activity and the potential mechanisms of resistance to BTK and PI3K inhibitors in aggressive B-cell lymphomas. We can find numerous publications on the B cell receptor signaling in B-cell lymphomas. However, the presented manuscript is not only a duplication of the existing ones. In my opinion, Profitos-Peleja et al. report in a well-organized way the recent developments in therapy with the use of inhibitors of BTK and PI3K. The authors summarize the ongoing clinical trials (Table 1) and provide a clear view of the effectiveness, options and side effects of recent therapeutic strategies.
Overall, this is a well-written, organized, and detailed manuscript. Table and figure improve the readability of the manuscript. They are clear and accurately reflect the conclusions presented in the text. The authors have adequately reviewed the medical literature. The references are appropriate. The paper is well written. The text is clear and easy to read.  

Author Response

(The authors gave the same response as above.)
